# The Narrowed Internal Auditory Canal: A Distinct Etiology of Pediatric Vestibular Paroxysmia

**DOI:** 10.3390/jcm11154300

**Published:** 2022-07-25

**Authors:** Samar A. Idriss, Hung Thai-Van, Riham Altaisan, Aicha Ltaief-Boudrigua, Pierre Reynard, Eugen Constant Ionescu

**Affiliations:** 1Department of Audiology and Otoneurological Explorations, Civil Hospitals of Lyon, 69003 Lyon, France; samar.a.idriss@hotmail.com (S.A.I.); hung.thai-van@chu-lyon.fr (H.T.-V.); pierre.reynard@chu-lyon.fr (P.R.); 2Department of Otolaryngology and Head and Neck Surgery, Holy Spirit University of Kaslik, Eye and Ear Hospital, Beirut 1201, Lebanon; 3Department of Audiology and Otoneurological Explorations, Claude Bernard Lyon 1 University, 69003 Lyon, France; 4Paris Hearing Institute, Institut Pasteur, Inserm U1120, 75015 Paris, France; 5Department of Otolaryngology Head and Neck Surgery, King Faisal University, Al-Ahsa 31982, Saudi Arabia; altaisan.riham@gmail.com; 6Department of Otolaryngology Head and Neck Surgery, CHU Besancon, 25056 Besancon, France; 7Department of Radiology, Civil Hospitals of Lyon, 69003 Lyon, France; aicha.ltaief-boudrigua@chu-lyon.fr

**Keywords:** narrowed internal auditory canal, neurovascular compression syndrome, vestibular paroxysmia, pediatric vertigo, anticonvulsant drugs, cochleovestibular nerve

## Abstract

Vestibular paroxysmia (VP) is a disorder encountered in the pediatric population that etiology has been attributed to neurovascular cross-compression syndrome (NVCC). The purpose of this study was to report a new probable pathological condition, the narrowed internal auditory canal (IAC), which appears to be involved in the development of a clinical picture of VP in the pediatric population. A retrospective descriptive comparative study was conducted to compare clinical, electrophysiological, radiological, and therapeutic outcomes in both etiologies. Overall, 16 pediatric patients suffering from VP were included and divided into two groups: patients with narrowed internal auditory (Group 1) were compared to those with NVCC syndrome (Group 2). Patients in both groups were similar in terms of auditory complaints, as well as hearing, vestibular, and electrophysiological status. A narrowed IAC was encountered in the adolescent age category and females, especially those with rapid growth. The diagnosis requires a careful analysis of the shape and diameters of the IAC. Radiologic measurements in the axial plane do not seem to be sufficient to confirm the diagnosis, and, therefore, an analysis of diameters in the coronal plane is required. Treatment with sodium-channel blockers drugs showed promising results not only by relieving vertigo but also by normalizing the electrophysiological findings. In conclusion, a narrowed IAC can be considered in patients suffering from VP.

## 1. Introduction

The prevalence of vertigo and balance impairment is estimated to be around 5–10% in the pediatric population [1,2,3]. Two peaks were reported by Brodsky et al.: the first one takes place during infancy and the second during adolescence [4]. In a recent search of the literature, Fancello et al. [5] identified vestibular migraine and migraine variants as the most common cause of vertigo (32.7%), followed by cochleo-vestibular disorders (23.9%). Vestibular paroxysmia (VP) is one vestibular disorder with unknown prevalence [6]. A definitive VP consists of at least ten recurrent, short (less than one minute), spontaneous attacks of stereotyped and self-limiting vertigo that respond to anticonvulsant drugs (carbamazepine/oxcarbazepine) [7]. Data on retrocochlear diseases in children are sparse [8,9], but neurovascular cross-compression (NVCC) syndrome has been described [10], and VP has been attributed to NVCC syndrome [11]. The pathophysiology of these attacks relies on the “ephaptic theory” [12]: an electrical conduction between the proximal part of the partially demyelinated eighth cranial nerve (CN VIII) and neighboring vascular structures of the cerebellopontine angle (CPA) [13]. In fact, the vascular structures of the CPA can cause a direct effect on the surrounding nerves either by direct segmental compression or by pulsatile vascular effect and consequently lead to a dysfunction of the vestibular and/or auditory nerves [7,14].

In addition to the diagnostic criteria of VP defined previously [7], several findings may be helpful to consider a retrocochlear origin of VP. The hyperventilation maneuver can modify the direction of an initial spontaneous nystagmus or cause a hyperventilation-induced nystagmus (HVIN) [15]. The latter has been reported to be more common in retrocochlear pathologies compared to inner ear end-organ pathologies [7,16]. Electrophysiologic modifications with absolute values of interpeak latencies (IPL) I–III exceeding 2.3 ms in patients with normal pure tone audiograms are a strong indication of CN VIII involvement [17]. Vestibular dysfunction can vary among patients [18,19], and caloric testing discloses mild increases of a vestibular deficit over time [20]. MRI with high-resolution T2 weighted sequence (DRIVE/CISS/FIESTA) of the brainstem can support the diagnosis [7]. The exclusive presence of a neurovascular contact is not considered to be sufficient. A perpendicular contact along two different perpendicular planes leading to a deviation of the nerve path and generation of a weak point of the CN via pressure is essential for the diagnosis [21]. VP in NVCC responds to low doses of fast sodium channel drugs such as carbamazepine (200–600 mg/day) or oxcarbazepine (300–900 mg/day); it has shown to be effective in children [14]. Although a microsurgical decompression can be proposed in refractory cases [14], only one pediatric case of CN VIII surgical decompression was reported in the literature and succeeded to reveal tinnitus in a 15-year-old female [22].

A narrowed internal auditory canal (IAC) is another retrocochlear condition that can occur concomitantly with cochleovestibular nerve hypoplasia or aplasia [23,24,25,26]. It has also been reported that narrowed IAC can generate a local nerve damage or neuropathy, hence generating cochleovestibular and/or facial symptoms [27,28,29,30]. In the present study, we compared clinical, electrophysiological, radiological, and therapeutic outcomes of two pediatric groups matching the diagnosis criteria of probable VP: those with NVCC syndrome to those with a narrowed IAC. The goal of this comparison is to elucidate the existence of a distinct pathologic condition that appears to be, along with NVCC syndrome, at the origin of VP in pediatric patients.

## 2. Methods

### 2.1. Population

A retrospective descriptive comparative study was conducted to compare two retrocochlear pathologies. Over a two-year period, 704 patients were referred to our tertiary care department, after a preliminary pediatric otolaryngology and/or neurological consultation, for evaluation of vertigo or related balance complaints. After a thorough vestibular evaluation, patients whose clinical presentation met the criteria of probable VP, as defined by the Bárány society [7], were included. These patients underwent a radiologic evaluation with a cerebral and inner ear MRI, and, only when narrowing of the IAC was suspected, a high-resolution computed tomography (HRCT) of temporal bones was subsequently realized to confirm the diagnosis (for more details, see radiology sections below). Subjects younger than 5 years old and older than 18 years old were ruled out. Cases with a personal or a family history of headache or migraine, as well as motion sickness, were excluded. Children who are candidates for cochlear implantation were excluded, as well. Furthermore, patients with confounding variables such as previous ototoxic medication consumption, middle ear disease, neurological disorders, ophthalmic or vergence anomalies, psychological status, or systemic diseases were excluded.

Overall, 32 children with a narrowed IAC were selected. A total of 24 patients were opted out because of confounding variables, incomplete medical records, or loss of follow-up, and 8 remaining patients were included (Group 1). A comparative group including the pediatric population with NVCC syndrome was randomly selected (Group 2).

The investigation adhered to the principles of the Declaration of Helsinki. Written informed consent was obtained from the children’s parents.

### 2.2. Cochleo-Vestibular Assessment

All children were subjected to cochleo-vestibular evaluation. A detailed medical and otological history was taken, and a complete otolaryngological and neurological examination was carried out by the same physician. Otoscopy and tympanometry were verified in each subject. The otoneurologic assessment included a battery of auditory and vestibular tests.

As part of an auditory evaluation, each child was subjected to a pure tone audiometry (PTA) and auditory brainstem-evoked responses (ABR). CN VIII involvement was suspected if the IPL I–III was prolonged to more than 2.3 ms on the interested side, as defined my Moller’s ABR criteria [17].

As part of vestibular evaluation each child was subjected to a videonystagmoscopy to look for spontaneous nystagmus, a positional nystagmus, and HVIN. Spontaneous nystagmus was considered pathologic when its mean slow-phase velocity exceeded four degree per second [31]. We also performed cervical vestibular evoked myogenic potentials (cVEMPs), a rotational chair testing to assess the vestibulo-ocular reflex (VOR), and a video head impulse test (vHIT). Therefore, otolithic function was assessed by cVEMPs elicited in bone conduction (BC), as previously described [32]. Although the utricle may also respond to BC stimuli, the presence of cVEMPs indicates predominantly human saccular response function [33]. Latencies and amplitudes of the first positive–negative peaks (p13–n23) were defined as described by Fujimoto et al. [34]. Absent response was defined by an a non-reproducible p13–n23 over two attempts, and decreased response was defined by a reproducible p13–n23 with an asymmetry ratio above the normal upper limits. For the rotational chair test, a rotational chair (Nagashima Co. Ltd., S-II, Tokyo Japan) was accelerated to a maximum rotational velocity of 160°/s and then decayed by 4°/s. The test was performed once in a clockwise direction and once in a counterclockwise direction. Eye movements were recorded by ENG, and the duration and number of beats of per rotatory nystagmus were calculated for the evaluation of semicircular canals [35]. Vestibular function was defined by gain, phase, and symmetry, as described by several pediatric studies, and normative values were settled according to the age group [36,37,38]. For vHIT, the Ulmer II system (Synapsys^®^, Marseille, France) was used, with two experienced right-handed examiners. This device is a widely recognized validated non-invasive tool to assess high-frequency VOR in children [39], and it is used in routine practice for vestibular deficit screening. Each SCC is sensitive to endolymph displacement according to its specific anatomic orientation and, hence, to acceleration in that plane (see Rabbitt’s mathematical model [40]). Five-to-ten acquisitions were made per SCC, beginning with the horizontal canal; examination time did not exceed 10 min. A normal response was defined by a gain value between 0.8 and 1.2, while a decrease in gain is defined by a value below 0.8 [41,42].

### 2.3. Radiological Assessment

All patients underwent a cerebral and inner ear 1.5 and/or 3 Tesla MRI to eliminate central pathologies and check for a cochleo-vestibular abnormality. Parasagittal images of the internal auditory canal demonstrate intact facial, vestibular, and cochlear nerves. The CN VIII course in the CPA and IAC was closely analyzed, and a sufficient nerve development was confirmed. When an NVCC syndrome was visualized with a normal IAC diameter, no further imaging was realized. Only when narrowing of the IAC was suspected did patients undergo an HRCT of temporal bones with measurements of the length, vertical diameter of the fundus (VDF), and vertical diameter of the porus (VDP) [43]. The normative measurements by CT among children, as defined by Marques et al., were as follows: length = 11.17 mm, VDF = 4.82 mm, and VDP = 7.53 [44]. In the present study, interpretations of the anteroposterior and craniocaudal diameters were assessed in both axial and coronal planes, respectively. A narrowed IAC was considered when an anteroposterior diameter measured less than 3 mm [44,45]. All the radiological analyses and measurements were carried out by the same radiologist.

### 2.4. Therapeutic Strategies

The efficacy of anticonvulsant drugs is one of the criteria that differentiate between a probable VP and a confirmed VP, among others [7]. In the present study, clinical observation was opted in patients with mild or moderate well-tolerated symptoms, and a probable diagnosis was defined. Treatment with anticonvulsant drugs was prescribed when symptoms were bothersome, recurrent, and incapacitating. The initial duration of treatment was set at six weeks, followed by a clinical evaluation at twelve weeks to assess the level and improvement.

## 3. Results

The demographic and clinical characteristics of both groups are shown in Table 1. The subjects’ age refers to the time of the first vertigo attack; thus, the first attack ranged between 5 and 18 years. In Groups 1 and 2, 87.5%, and 50% of subjects were adolescents (≥10 years-old), respectively. Females were more commonly concerned: 75% in Group 1, and 62.5% in Group 2.

### 3.1. Clinical Data

In addition to VP, two patients (25%) with narrowed IAC described an effort-induced vertigo, and only one patient (12.5%) had retro-auricular pain. In patients with NVCC syndrome, imbalance was frequently mentioned (50%), followed by retro-auricular pain (25%).

Overall, although hypoacusis was not a common subjective complaint, hearing loss was more frequently documented. Two patients had sensorineural hearing loss (SNHL) in each group. While HL was unilateral in 50% of cases in Group 1, all cases of HL were unilateral in Group 2. Tinnitus was described in one patient in each group; both patients had normal hearing status.

On videonystagmoscopy, a spontaneous nystagmus was detected in 62.5% and 37.5% in Groups 1 and 2, respectively. Only one patient (Group 1) had spontaneous vertical nystagmus that could suggest a central cause, but it was well inhibited by fixation. The HVIN turned out to be positive in 66% of cases (three patients) for each group.

### 3.2. Cochleo-Vestibular Data

Cochleo-vestibular data for each group are described in Table 2. IPL I–III latencies were prolonged in eight ears in narrowed IAC (50%) and five ears (30%) in NVCC syndrome. Vestibular outcomes were variable. While assessing otolithic function via VEMPs, no dysfunction was documented in Group 1, and unilateral dysfunction was documented in 50% of cases in Group 2. When assessing the semicircular canals’ function, Group 1 seemed to have decreased VOR in 37.5% of cases. In Group 2, the VOR deficit was evoked in two patients but was not coherent with the laterality of the vascular loop. Caloric testing, with a cutoff level 20% for caloric weakness [46], was realized among six patients in Group 1 and two patients in Group 2. While it was normal in 66% of cases in Group 1, it showed a deficit in 50% of cases in Group 2 which was ipsilateral to the NVCC. A decreased gain on VHIT was documented in 37.5% and 25% of cases in Group 1 and 2, respectively.

### 3.3. Radiologic Data

Table 3 shows detailed radiological interpretations and measurements for both groups. While the neurovascular compression was unilateral in all cases in Group 2, the IAC narrowing was bilateral in all cases in Group 1, except for one case. A right-sided NVCC was more frequently encountered (62.5%) (see Appendix A).

### 3.4. Therapeutic Data

Treatment with anticonvulsant drugs was prescribed to alleviate severe and recurrent incapacitating attacks [7,10]. Therefore, oxcarbazepine was prescribed in four patients for each group. All patients reported substantial improvement consistent with the onset of drug use, except for one patient in Group 2. Half of these patients described the reappearance of VP when treatment was withdrawn in the short term. In these cases, treatment was continued for longer periods.

### 3.5. Case Report

An 18-year-old girl has been suffering from recurrent attacks of vertigo consistent with paroxysmal vertigo for more than 2 years. She did not have any cochlear symptoms, and her audiometry was normal. Her examination did not reveal a spontaneous or an induced nystagmus. Her cochleovestibular testing showed a normal otolithic function, symmetrical caloric responses, and normal gains on vHIT. On the rotational chair, her VOR was diminished. The ABR revealed a slight increase of the left-sided IPL I–III (2.37 Vs 2.27 for the right ear). When she underwent an MRI of the brain and inner ear, a narrowing of the IAC was suspected bilaterally, more specifically on coronal planes and more pronounced on the right (Figure 1).

Consequently, an HRCT was performed. In the coronal plane, the smallest craniocaudal diameter was estimated at 2.3 and 2.7 mm on the right and left side, respectively (Figure 2).

In view of the recurrence and the disability of vertigo, preventing the patient from continuing her daily activity, a treatment with oxcarbazepine was prescribed at a dose of 300 mg. During treatment, the patient described a significant reduction of attacks. The IPL I–III was also normalized under treatment: 1.87 ms for the right ear and 2.07 ms for the left ear (Figure 3). Withdrawal of treatment was associated with a recurrence of symptoms, and this necessitated prolonging its duration.

## 4. Discussion

### 4.1. The Narrowed Internal Auditory Canal

Our main objective of this paper is to emphasize a new etiology that can be considered among pediatric patients with VP. Our data suggest patients with narrowed IAC had similar clinical manifestations and cochleo-vestibular outcomes to those with NVCC. The IAC is a bony canal of the temporal bone that involves the CN VII (facial), the CN VIII (vestibulocochlear), and the labyrinthine artery, a branch of the anterior inferior cerebellar artery [47]. As stated earlier, it is assumed that a local compression of CN VIII can generate VP [29,48]. Furthermore, in the case of a narrowed IAC, we may assume that the cochleo-vestibular symptoms can be related to an entrapment-type local neuropathy similar to the pathology of the carpal tunnel or radiculopathies caused by the local compression [49,50]. In fact, congenital IAC stenosis can be responsible of various cochleo-vestibular and facial symptoms, but it is often associated with hypoplastic nerve [23,51,52]. In our study, radiologic evaluation confirmed sufficient nerve development, but a narrowed IAC. Previously, one case study reported cochleovestibular and facial nerve dysfunction secondary to IAC stenosis [53]. All of these data suggest that CN VIII compression, secondary to a small diameter of the IAC, may lead to local nerve damage, resulting in a possible ectopic excitation or inhibition of the involved cranial nerve fibers, as previously reported [54]. Therefore, a narrowed IAC might be considered in patients presenting a VP associated or not to auditory symptoms.

### 4.2. Morphometric Considerations

In the present study, our data suggested that a narrowed IAC is more commonly encountered in adolescence, as compared to childhood. Surprisingly, most of these adolescents had a noteworthy height growth (data not shown). On the one hand, growth happens in a craniocaudal direction with downward slanting of nerve roots [55]. On the other hand, the size and the shape of the IAC vary considerably among individuals; the wall length and porus diameter increase with age, but the diameter of the fundus remains stable [43,56]. The diameter and length increase significantly until the age of 1 year and 10 years, respectively [56]. A rapid height growth may be associated with an increased growth of the IAC length and consequently generate a nerve stretching, excitement, or damage of the nerves. In fact, an extensive development of a nerve tends to delay its neighbors, and a lack of development tends to excite them to a more active growth [57].

Another morphological criterion concerns the shape of the IAC, as it is not unique and can take several forms [44]. Hence, radiologic measurements, including opening width, longitudinal length, and vertical diameter, can be sufficient when the IAC is cylindrical. However, when IAC has a conical or a bud shape, these measurements may be insufficient, and one vertical measurement might be biased. Therefore, we elected to verify the smallest vertical diameter in addition to conventional dimensions in both the axial and coronal planes to avoid misdiagnosis. In fact, the association between otological symptoms and vascular loop are still controversial [58,59,60,61,62]. This relationship has been related to the loop location following Chavda classification [63]. While Yoo et al. reported that tinnitus was significantly higher when the loop lies within the cerebellopontine angle (Type I) and when the loop is extended within less than 50% of the IAC (Type II) [59]. Kim et al. found that tinnitus was significantly related to loops’ extension more than 50% (Type III) [60]. Hearing loss was significantly associated with loops running between the CN VII and CN VIII [61]. Taken together, these data suggest that a narrowing of the shape-related smallest diameter of the IAC may generate cochleo-vestibular symptoms. It would be interesting to radiologically explore an associated lack of the protective property of the cerebrospinal fluid at the level of the suspected compression suggestive of a mechanical injury [64]. A grading of the internal auditory canal assessment, as per lumbar spinal stenosis [65], would strengthen radiologic practice and create a unified view of analysis and reporting system.

### 4.3. Therapeutic Considerations

In the present study, low doses of oxcarbazepine were prescribed for patients with invalidating symptoms, and it succeeded to relieve patients’ symptoms. Oxcarbazepine’s efficacy can somehow be secondary to its ability to bind to sodium channels and inhibit repetitive neuronal exciting firing [66]. Moreover, oxcarbazepine can inhibit glutamate, which is the most important afferent neurotransmitter in the auditory system [66,67]. So far, there are no randomized controlled trials for VP, but the response to treatment with carbamazepine/oxcarbazepine supports the diagnosis [14]. Although the long-term administration of anticonvulsant drugs in epileptic patients was reported to provoke delayed auditory conduction [68,69,70], a normalization of IPL I–III was noticed in our case. Thus, it is possible that the low doses of oxcarbazepine used in VP are less toxic than the higher doses usually required in epilepsy, resulting in a beneficial effect on the narrowed IAC-induced neuropathy. Unfortunately, we were unable to control electrophysiological outcomes both pretreatment and per treatment in all patients; thus, further studies are required to support the efficacy of treatment in such pathologies. A conduction latencies modification needs to be evaluated in a larger population sample for its usefulness as a possible additional objective criterion to reinforce the diagnosis and evaluate the treatment’s effectiveness.

### 4.4. Limitations and Further Research

In the present study, we report a distinct possible etiology that can be considered as the cause of VP in children. The results of this study should be considered in light of certain limitations. The sample size was small (n = 16), so it is too insufficient to obtain statistical data and to be generalized to the pediatric population. The results’ estimates are based on retrospective observational data. They are therefore subject to bias. In addition, therapeutic data were scarce. Thus, a clear judgment on therapeutic response was missing. This topic, for which, to our knowledge, no previous studies have been conducted, requires further research. Additional studies with larger sample size, statistical data, therapeutic response, and long-term evaluations may be useful to consider the diagnosis broadly and generalize it to the pediatric population.

## 5. Conclusions

VP is a documented etiology of vertigo in the pediatric population and has been related to NVCC syndrome. In the present study, our data suggest that VP can also originate from a narrowed IAC. Clinical presentation and electrophysiological outcomes can be similar to those of NVCC syndrome. The diagnosis is suspected through MRI and can be confirmed by a supplementary HRCT of temporal bones with analysis of the shape, diameter, and opening width of the IAC in both axial and coronal plans. In the present study, we reported a systematic association between the narrowing of IAC diameter, especially in the coronal plane (less than 3 mm) and VP symptomatology, thus suggesting its involvement in the genesis of this pathology. This finding urges the concerned specialists to carefully assess the IAC diameter, not only in the axial plane, but also in the coronal plane. The latter can be evaluated initially on an MRI and, if deemed necessary, completed with a CT scan. Treatment with sodium-channel blockers drugs showed promising results. Although the present study is limited by a small sample size, it allows clinicians to become sensibilized to this VP’s etiology. Further studies need to be conducted to assess the characteristics of this new clinical entity.

## Figures and Tables

**Figure 1 jcm-11-04300-f001:**
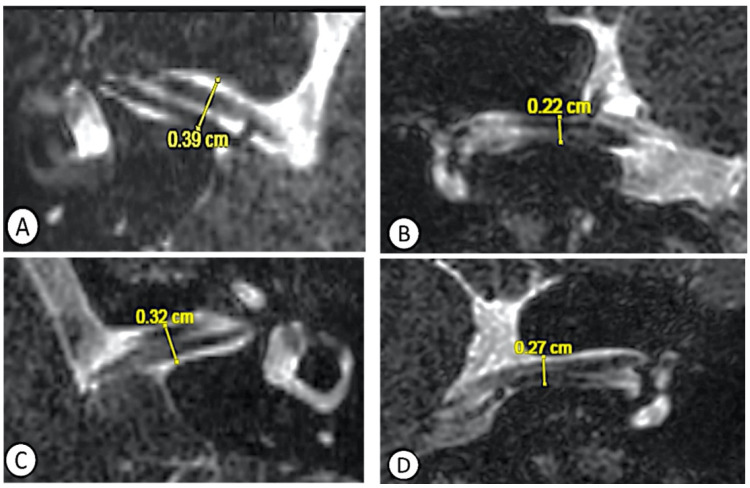
MRI of the inner ear (IAC sections) suggesting a bilateral narrowing of the IAC in the coronal plane, more pronounced on the right side: (**A**) right ear, axial plane; (**B**) right ear, coronal plane; (**C**) left ear, axial plane; and (**D**) left ear, coronal plane.

**Figure 2 jcm-11-04300-f002:**
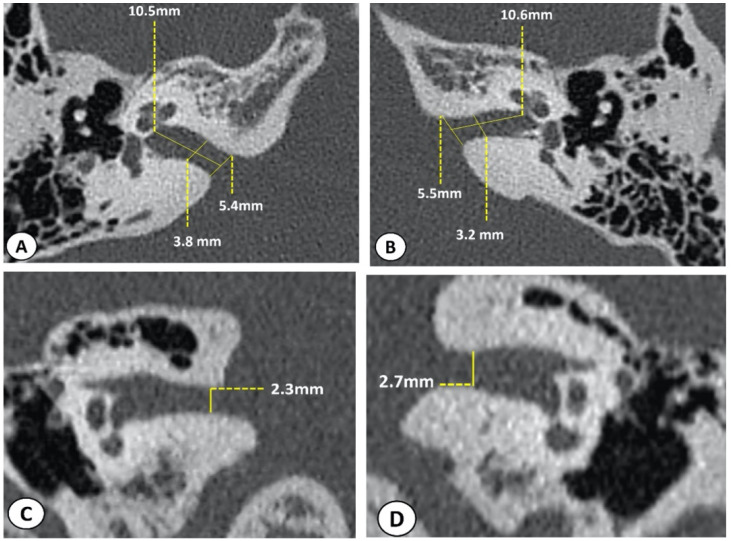
HRCT of temporal bones (IAC sections) confirming a bilateral narrowing of the IAC, in the coronal plane, more pronounced on the right side: (**A**) right ear, axial plane; (**B**) left ear, axial plane; (**C**) right ear, coronal plane; and (**D**) left ear, coronal plane.

**Figure 3 jcm-11-04300-f003:**
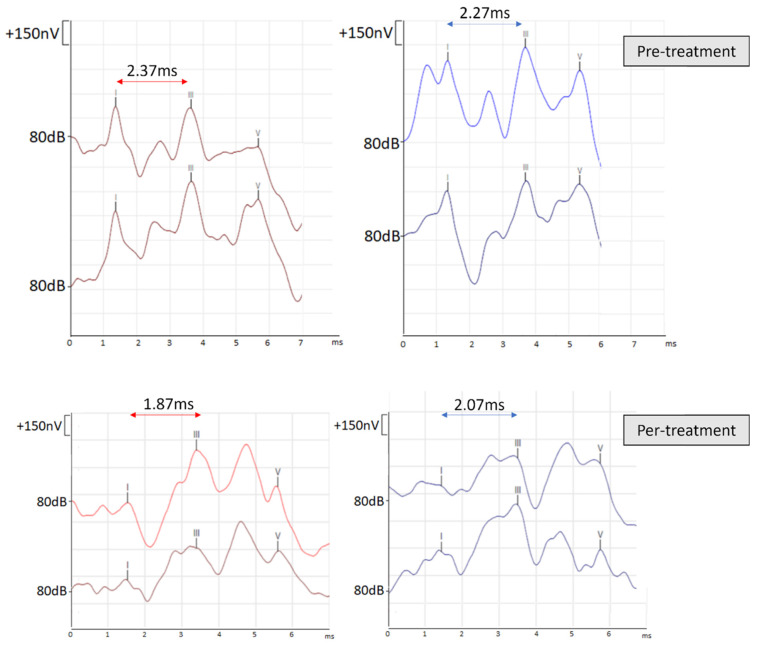
ABR and IPL I–III measures pretreatment and per treatment.

**Table 1 jcm-11-04300-t001:** Demographic and clinical data for Groups 1, 2, and 3. Abbreviations: F (female), HL (Hearing loss), L (left), M (male), and R (right), VP (Vestibular paroxysmia).

	Sex	Age (Years)	Vestibular Symptoms (Other than VP)	Auditory Symptoms	Pure Tone Audiogram	Spontaneous Nystagmus	Hyperventilation Maneuver
**Group 1**							
1	M	8	None	Absent	N	Absent	Absent
2	F	12	None	Absent	N	(L) horizontal	Absent
3	F	15	None	(R) Hypoacusis	(R) SNHL	(R) horizontal	Absent
4	M	15	Effort-induced vertigo	Absent	N	(R) superior	(R) inferior
5	F	16	Positional vertigo Effort-induced vertigo (L) Retro auricular pain	Absent	N	Absent	Absent
6	F	16	None	Absent	N	Absent	Absent
7	F	17	None	Absent	(x2) moderate HL	(R) horizontal	(L) Nystagmus
8	F	18	None	Bilateral tinnitus	N	(L) horizontal	(R) horizontal
**Group 2**							
1	M	5	Imbalance	Absent	(L) severe HL	Absent	Absent
2	F	6	Imbalance	Absent	N	Absent	Absent
3	M	6	None	Absent	N	Absent	Absent
4	F	8	Imbalance	Bilateral tinnitus	N	(R) horizontal	Absent
5	F	12	(L) Retro auricular pain	Absent	N	(R) horizontal	Accentuation of nystagmus
6	F	13	Imbalance	Absent	N	(R) horizontal	Accentuation of nystagmus
7	M	14	None	Absent	N	Absent	(L) nystagmus
8	F	16	Retro auricular pain	(L) fluctuating	(L) moderate HL	Absent	Absent

**Table 2 jcm-11-04300-t002:** Audio vestibular evaluation for Groups 1, 2, and 3. Abbreviations: ABR (auditory brainstem response), AC (anterior canal), L (left), LC (lateral canal), PC (posterior canal), R (right), VEMPs (vestibular evoked myogenic potentials), vHIT (video head impulse test) and VOR (vestibulo-ocular reflex).

	ABR I–III Interval		VOR	vHIT	VEMPs
	(R)	(L)			
**Group 1**					
1	2.12	2.28	N	Decreased gain (x2)	N
2	2.2	2.3	(R) decreased responses	N	N
3	2.42	2.46	N	Decreased gain PC (x2)	N
4	2.37	2.46	Bilateral decreased responses	N	N
5	2.27	2.37	N	N	N
6	2.5	2.5	N	N	N
7	2.04	No response	N	Decreased gain LAC & LLC	N
8	1.96	1.92	Bilateral decreased responses	N	N
**Group 2**					
1	1.83	2.25	N	Decreased gain LPC	Absent (L)
2	2.2	2.37	(R) decreased responses	N	Absent (R)
3	2.20	2.20	N	N	Absent (R)
4	2.50	2.10	N	N	Absent (R)
5	2.21	2.33	N	Decreased gain RLC	N
6	2.07	2.10	N	N	N
7	2.23	2.23	(L) decreased responses	N	N
8	2.36	2.36	N	N	N

**Table 3 jcm-11-04300-t003:** Radiologic data and IAC measurements (in mm). AP (anteroposterior), L (length), CC (craniocaudal), H (height), MRI (magnetic resonance imaging), and VDP (vertical diameter of the porus).

Group 1				
	MRI	CT Scan of Temporal Bones + IAC Measurements (mm)		Conclusion
		Right	Left	
**1**	The internal auditory canals are small in size bilaterally.	- Axial plane: VDP: 7.2 Smallest AP diameter: 3.3 L: 10.9 - Coronal plane: Smallest CC diameter: 2.8	- Axial plane: VDP: 8.6 Smallest AP diameter: 2.9 L: 11.3 - Coronal plane: Smallest CC diameter: 2.2	Bilateral narrowing of the CC caliber of IACs, more marked on the left side.
**2**	The internal auditory canals are small in size bilaterally.	- Axial plane: VDP: 4.2 Smallest AP diameter: 4 L: 12.9 - Coronal plane: Smallest CC diameter: 2.3	- Axial plane: VDP: 5.8 Smallest AP diameter: 3.5 L: 13.1 - Coronal plane: Smallest CC diameter: 2.7	Bilateral narrowing of the CC caliber of IACs, more marked on the right side.
**3**	Bifid aspect of the left saccule. The internal auditory canals are small in size bilaterally.	- Axial plane: VDP: 4.6 Smallest AP diameter: 3.5 L: 10.9 - Coronal plane: Smallest CC diameter: 2	- Axial plane: VDP: 5.3 Smallest AP diameter: 4.3 L: 5.3 - Coronal plane: Smallest CC diameter: 3.4	Right narowing of the CC caliber of the IACs.
**4**	The internal auditory canals are small in size bilaterally.	- Axial plane: VDP: 4.6 Smallest AP diameter: 3.5 L: 13.7 - Coronal plane: Smallest CC diameter: 2.3	- Axial plane: VDP: 4.8 Smallest diameter: 4.1 L: 13 - Coronal plane: Smallest CC diameter: 2.3	Bilateral narrowing of the CC caliber of the IACs.
**5**	Narrowing of the IAC in their CC caliber more marked on the right.	- Axial plane: VDP: 5.4 Smallest AP diameter: 3.8 L: 10.5 - Coronal plane: Smallest CC diameter: 2.3	- Axial plane: VDP: 5.5 Smallest AP diameter 3.2 L: 10.6 - Coronal plane: Smallest CC diameter: 2.7	Bilateral narrowing of the CC caliber of the IACs, more marked on the right side.
**6**	The internal auditory canals are small in size bilaterally.	- Axial plane: VDP: 4.7 Smallest AP diameter: 4.1 L: 15.7 - Coronal plane: Smallest CC diameter: 1.5	- Axial plane: VDP: 8 Smallest AP diameter: 3.6 L: 16.3 - Coronal plane: Smallest CC diameter: 2.5	Bilateral narrowing of the CC caliber of IACs, more marked on the right side.
**7**	Bilateral IAC narrowing	- Axial plane: VDP: 4.5 Smallest AP diameter: 4.1 L: 12.4 - Coronal plane: Smallest CC diameter: 1.8	- Axial plane: VDP: 4.7 Smallest AP diameter: 3.4 L: 14.8 - Coronal plane: Smallest CC diameter: 2.3	Bilateral narrowing of the CC caliber of the IACs, more marked on the right side.
**8**	The internal auditory canals are small in size bilaterally.	- Axial plane: VDP: 3.2 Smallest AP diameter: 2.7 L: 9.7 - Coronal plane: Smallest CC diameter: 2.1	- Axial plane: VDP: 8.4 Smallest AP diameter: 2.5 L: 10.7 - Coronal plane: Smallest CC diameter: 1.8	Bilateral narrowing of the CC caliber of the IACs, more marked on the left side.
**Group 2**				
**MRI**				
1	Left PICA vascular loop in contact with the cisternal emergence of the left acoustic facial bundle.			
2	Vascular loop at the level of the left APC; close contact between the left AICA and the acoustic-facial bundle at this level.			
3	Vascular loop at the level of the right APC			
4	Arterial vascular loop of the right AICA having a double contact with the cisternal path of the right nerve VIII, especially at the porus, in RE Z zone, without deviation of the neural structure.			
5	Double contact between an AICA at the emergence of the VIII on the right.			
6	Neurovascular compression syndrome on the right-sided bundle of NC VIII with a moderate mass effect on the nerve structure (pathogenic appearance).			
7	Intimate contact of the APC between VII and VIII with right PICA.			
8	Vascular-nerve contact between the left AICA and the left acoustic-facial bundle at the level of the IAC porus. This contact is orthogonal with the G facial nerve and VIII G. Mass effect on VIII questionable in transitional zone.			

## Data Availability

Not applicable.

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
