# Peer review of "The Narrowed Internal Auditory Canal: A Distinct Etiology of Pediatric Vestibular Paroxysmia"

_jcm, 2022, doi:10.3390/jcm11154300_

Round 1

Reviewer 1 Report

Reviewer Comments

General Comments

 1.       Be mindful there are different fonts throughout the text

Specific Comments

 Methods

1.       Population:  the authors specify that 704 patients attended the clinic but do not list what symptoms they presented with, this should be included at this stage.  In general the description of the patient population is unclear.

2.      The authors then suggest that they selected 32 patients from that group (who presumably had the presenting symptom of dizziness who had a narrowed IAC.  So of course there will be a correlation between the presenting symptom ? dizziness and the radiologic finding because they were selected as such.  They do describe a comparator group but do not and should include a control group (i.e. a population without the symptom but who may also have the radiologic finding as this may be considered a normal variant.

Results

1.      In the description of the radiologic findings the authors do not outline whether or not the imaging was examined for the presence of the typical 4 nerves in the IAC typical detected in the parasagittal plane. This should be included.

Discussion

1.      The authors make the statement that “ it has been reported that a narrowed IAC was associated with ipsi SNHL” and state a couple of papers.  What they have not included but should be is that the narrow IAC is a marker for absence of the cochlear nerve, so this is not a fair comparison.

Author Response

JCM-1806037

Reviewer 1

On behalf of the manuscript authors, I would like to thank you for your effort in reviewing the manuscript and I appreciate your comments.

General Comments

  1. Be mindful there are different fonts throughout the text Specific.

The font has been unified for the whole text.

Methods

  1. Population: the authors specify that 704 patients attended the clinic but do not list what symptoms they presented with, this should be included at this stage. In general, the description of the patient population is unclear.

  1. The authors then suggest that they selected 32 patients from that group (who presumably had the presenting symptom of dizziness who had a narrowed IAC. So of course, there will be a correlation between the presenting symptoms? dizziness and the radiologic finding because they were selected as such. They do describe a comparator group but do not and should include a control group (i.e., a population without the symptom but who may also have the radiologic finding as this may be considered a normal variant.

We apologize if our original inclusion criteria were unclear, we have modified them to be clearer [Lines 101-110].

In the present study, the primary inclusion criteria were patients who met the criteria of a vestibular paroxysmia (as defined by the Bárány society). These patients underwent MRI to eliminate a central or peripheral pathology at the origin of the patients’ symptoms. An analysis of the IAC was realized and only when narrowing of IAC narrowing was suspected. We ended up with 32 patients who had a confirmed diagnosis with a narrowed IAC.

To our knowledge, the causality of the narrowed IAC pathology to vestibular paroxysmia was not described elsewhere, contrary to NVCC. Therefore, the present study was conducted.

We appreciate the reviewer’s insightful suggestion and agree that it would be useful to demonstrate to present a control group where subjects have VP without a retrocochlear disease. However, in view of previous studies reporting on VP (Hufner K et al. 2008, Best C et al. 2013, among others), NVCC was highly prevalent among patients. For this reason, we decided to compare two different etiologies for one single disease without a control “healthy” group. Also, an asymptomatic control group is less likely to be feasible since diagnosis requires a HRCT of the temporal bone. The latter is considered an invasive test and cannot be performed in pediatric populations without reasonable indication.   

Results

  1. In the description of the radiologic findings the authors do not outline whether or not the imaging was examined for the presence of the typical 4 nerves in the IAC typical detected in the parasagittal plane. This should be included.

This observation is correct. We have added the requested information in the text [Line 162-165].

Discussion

  1. The authors make the statement that “it has been reported that a narrowed IAC was associated with ipsi SNHL” and state a couple of papers. What they have not included but should be is that the narrow IAC is a marker for absence of the cochlear nerve, so this is not a fair comparison.

We thank the reviewer for pointing this out. We have removed the noted sentence since the absence of cochlear nerve was already mentioned [Lines 285-286].

Reviewer 2 Report

The manuscript is well written, and the argument sounds relevant. The author presents a new possible observation of clinical relevance, in terms of an anatomic condition (narrowed internal auditory canal) giving symptoms mimicking vestibular paroxysmia. My concern is about the quantification of different test results, without any mention of normative data or sources to these. It may sound more qualitative if the test normative data/limit for pathological results with confidence interval will be added. Other minor concerns both factual and formal should be also taken up. I attach my comments in the PDF. 

Author Response

On behalf of the manuscript authors, I would like to thank you for your effort in reviewing the manuscript and I appreciate your comments.

Reviewer 3 Report

Review

to manuscript “The narrowed Internal Auditory Canal: a neglected cause of pediatric vestibular paroxysmia”

Samar A. Idriss, Hung Thai-Van, Riham Altaisan, Aicha Ltaief-Boudrigua, Pierre Reynard and Eugen Ionescu

In this study authors report about neglected cause of pediatric vestibular paroxysmia (VP) (which etiology has been attributed to neurovascular cross-compression syndrome - NVCC) associated with narrowed internal auditory canal. For investigate association between VP and narrow internal auditory canal authors applied a retrospective descriptive comparative study was conducted to compare clinical, electrophysiological, radiological, and therapeutic outcomes.

In total 16 pediatric patients suffering from VP were included and divided in two groups; patients with narrowed internal auditory (Group 1) were compared to those with NVCC syndrome (Group 2). Patients with both groups were similar in terms of auditory complaints, as well as hearing, vestibular and electrophysiological status. In conclusion, authors suggest a considered the radiological data about narrowed IAC in patients suffering from VP.

In overall this is a good comparative job. I really enjoyed reading it. However, I have some comments and recommendations.

Major comments:

1. In my opinion, the size of IAC is a very indirect morphology sign. In this case, the association between radiological data and clinical findings is very a difficult objective. Although, theoretically, suggested hypothesis is most likely correct. Maybe I'm wrong, but I think that random Group 2 is not clearly correct for comparative analyses. For evidence of the narrowed IAC and VP comparative group should be present from control individuals without retrocochlear dysfunction.

Also, in this paper, present only empiric data with subjective evaluation, statistical analysis between the two studied groups of patients is absent.

Taking together, based on the existing design of the study, I suggest the following recommendation:

- Authors should be careful with conclusions and with terms. In this case, authors should change the title, perhaps in a questioning tone and generally correct the affirmative wording about association between IAC and VP throughout the text of the manuscript.

2. The case report presented in the Result section looks weird, this case not discussed in other sections. I think, for more objectives, the right decision would be demonstrate the all 8 cases with narrowed IAC in supplementary data.

Minor comment:

1. In my opinion, in this manuscript the section about limitation of study is missing. Also, I suggest added in this, or in the discussion section, the plans to expand of the research. 

Recommendation

Reconsider after major revision

Author Response

(The authors gave the same response as above.)

Round 2

Reviewer 3 Report

No comments. Good Luck!